# Electrical control of spins and giant *g*-factors in ring-like coupled quantum dots

H. Potts [1,3]*, I.–J. Chen[1,3], A. Tsintzis[1], M. Nilsson [1], S. Lehmann [1], K.A. Dick[1,2], M. Leijnse[1] & C. Thelander[1]*

Emerging theoretical concepts for quantum technologies have driven a continuous search for structures where a quantum state, such as spin, can be manipulated efficiently. Central to many concepts is the ability to control a system by electric and magnetic fields, relying on strong spin-orbit interaction and a large *g*-factor. Here, we present a mechanism for spin and orbital manipulation using small electric and magnetic fields. By hybridizing specific quantum dot states at two points inside InAs nanowires, nearly perfect quantum rings form. Large and highly anisotropic effective *g*-factors are observed, explained by a strong orbital contribution. Importantly, we find that the orbital contributions can be efficiently quenched by simply detuning the individual quantum dot levels with an electric field. In this way, we demonstrate not only control of the effective *g*-factor from 80 to almost 0 for the same charge state, but also electrostatic change of the ground state spin.

[1] Division of Solid State Physics and NanoLund, Lund University, SE-221 00 Lund, Sweden. [2] Centre for Analysis and Synthesis, Lund University, SE-221 00 Lund, Sweden. [3] These authors contributed equally: H. Potts, I.-J. Chen. *email: heidi.potts@ftf.lth.se; claes.thelander@ftf.lth.se

C ompared to real atoms, quantum dots (QDs) offer considerable advantages in the manipulation of their properties through applied electric and magnetic fields[1]. Importantly, ground state transitions and various level interactions in QDs can be studied using accessible magnetic fields, primarily explained by their large size and small energy scales. Development of two-dimensional semiconductor heterostructures provided a particularly flexible platform to study the orbital structure of QDs with various symmetries[1,2], as well as of molecular states resulting from the coupling of QDs[3,4]. Another milestone in QD research was the discovery of carbon nanotubes, a unique material with exceptionally strong confinement, and a special cylindrical geometry with consequences for spin and orbital interactions[5–8]. QDs made from carbon nanotubes were found to have a very different electronic structure compared to conventional QDs, with a nearly fourfold orbital- and spin-degeneracy, broken by spin-orbit interaction[7,9,10], and anisotropic effective g-factors ($g^*$) resulting from the orbital contributions[5].

Nanowire synthesis further expanded the prospects for controlling orbital and spin states by providing access to narrow bandgap materials, such as InAs and InSb, with inherent strong spin-orbit coupling and confinement effects[11,12]. These structures have now become cornerstones in Majorana research[13–15] and have been considered as a basis for spin qubits. Strongly level dependent $g^*$ have here been found, with values that can be many times higher than their bulk counterparts[11,12,15]. Recent theory work has explained the anomalous g-factors in many-electron nanowire QDs by a significant orbital contribution to the state[16]. Electrostatic tuning of these large g-factors would provide additional possibilities to achieve strong coupling of spins and cavity photons[17]. However, so far tuning of $g^*$ is limited to a relatively small range[18–20], and often relies on changing the charge state[11,12].

Here we explore the physics of strongly confined double-QDs inside InAs nanowires, having two connection points instead of just one. We find that some interacting orbitals exhibit vanishing hybridization energies, and an electronic structure almost identical to carbon nanotube QDs, with highly anisotropic $|g^*|$ varying from 3 to 80. However, in contrast to carbon nanotubes, it is possible to quench the orbital contribution using electric fields, and control $|g^*|$ from ~80 to ~0 for the same charge state. Using perturbation theory, we confirm that double-QD rings can form when an odd orbital on one QD hybridizes with an even orbital on the other QD.

## Results

**Coupling strongly confined quantum dots.** The quantum rings studied in this work appear inside quantum wells (QWs) formed during epitaxial growth of InAs nanowires[21]. The nanowire crystal phase is controlled to form structures similar to that shown in Fig. 1a, having a 5 nm long zinc-blende (ZB) segment sandwiched between two wurtzite (WZ) segments of about 30 nm. Due to a conduction-band offset, the WZ segments confine electrons in the centre ZB segment, resulting in a thin QW accessed by two tunnel barriers. We study the properties of the QW by first fabricating source-, drain- and gate-electrodes, as indicated in Fig. 1b, followed by transport measurements in a dilution refrigerator equipped with a vector magnet.

The electrostatics in the QW is manipulated through voltages applied to two side-gates ($V_L$ and $V_R$) and a global back-gate ($V_{BG}$), such that two QDs form, which are parallel-coupled to source and drain (Fig. 1d). The existence of a double QD is evidenced by the characteristic honeycomb diagram shown in Fig. 1e. Here, the strong confinement allows extraction of the

electron population on each QD, and provides a clear spin-pairing of orbital states, such that the orbital number ($O_L$, $O_R$) can be extracted. For this system, QD occupancies and inter-dot tunnel coupling are tunable over a wide range owing to the relatively rigid crystal phase confinement[22].

Previous works on these structures focused on the first orbital crossing ($O_{L,R} = 1$), for which each QD contains zero to two electrons[22–24]. In this work, we instead investigate interactions of higher orbitals. The crossing of ($O_L$, $O_R$) = (2, 3) (highlighted in Fig. 1e) is treated in the main text, whereas results from other crossings (both from the same and another device) are shown in the Supplementary Figs. 1–7. The electron population on each QD for this crossing is indicated in Fig. 1f. However, since filled orbitals are considered not to interact with other electrons, we will instead refer to the one- two- and three-electron regime (1e, 2e, 3e) in the rest of the article. We note that the conductance lines outlining the (2,3) crossing show very sharp corners. If interpreted in a standard double-QD picture, this would imply the absence of inter-dot tunnel coupling, which is in stark contrast to the clear hybridization gap (rounded corners) of the neighboring (2,2) crossing.

Transport is investigated by recording differential conductance, $dI/dV_{ds}$, as a function of drain-source voltage ($V_{ds}$) and either $V_{L,R}$ or magnetic field (**B**). The $V_{L,R}$ vectors are represented as green, red and yellow vectors in Fig. 1f, while the direction of the **B**-field with respect to the nanowire is defined in Fig. 1c. A measurement recorded along the green gate vector is shown in Fig. 1g for $B_{||} = 0.05$ T aligned with the nanowire. Sequential tunnelling here provides weakly outlined Coulomb diamonds representing the 1e, 2e and 3e regime. Within the Coulomb diamonds, the electron number in the QD system is constant, and the onset of co-tunnelling processes allows extraction of excited state energies with respect to the ground state. In the 1e and 3e regime, the 1st excited state corresponds to a **B**-field induced splitting of the ground state. Effective $|g^*|$-factors of ~64 and 40 can be extracted from $E_Z = |g^*|\mu_B B$ for the 1e and 3e regime, respectively. These values are significantly larger than for bulk InAs (−14.9), which indicates a very strong orbital contribution[11,16].

**Forming a quantum ring.** To understand the orbital contribution to $g^*$ we first study the evolution of states as functions of **B**-field strength and direction. We start by investigating the 1e regime at a $V_{L,R}$ where $O_L = 2$ and $O_R = 3$ are degenerate (referred to as zero detuning, $\Delta_{orb} = 0$). Figure 2a shows transport as a function of $V_{ds}$ and $B_{||}$. Three excited states evolving with different slopes can be identified, indicating that the underlying process is not just a standard Zeeman splitting of spin states.

In order to understand this behaviour, we implemented a three-dimensional simulation of the device structure, including a Rashba-type spin-orbit interaction term. By solving the single-electron Schrödinger equation, we can extract the energies of the ground state (GS) and the excited states (ESs). Figure 2b shows the states in the 1e regime at the crossing of the 2nd and 3rd orbitals. Hybridization of these orbitals leads to four new states, two of which increase rapidly in energy with $B_{||}$, and two of which decrease. This can be understood in the following way: The hybridized wave function resembles a ring in which electrons can orbit clockwise or anti-clockwise (blue/red), similar to what is known from carbon nanotube QDs[7,9]. Fig. 2c shows calculated onsets of inelastic co-tunnelling processes through excited states, which can be directly compared to our experimental results. Excellent quantitative agreement was obtained by adjusting the nanowire surface charge, the effective spin g-factor ($g^*_{spin}$), and the SOI strength, while keeping these within the ranges of

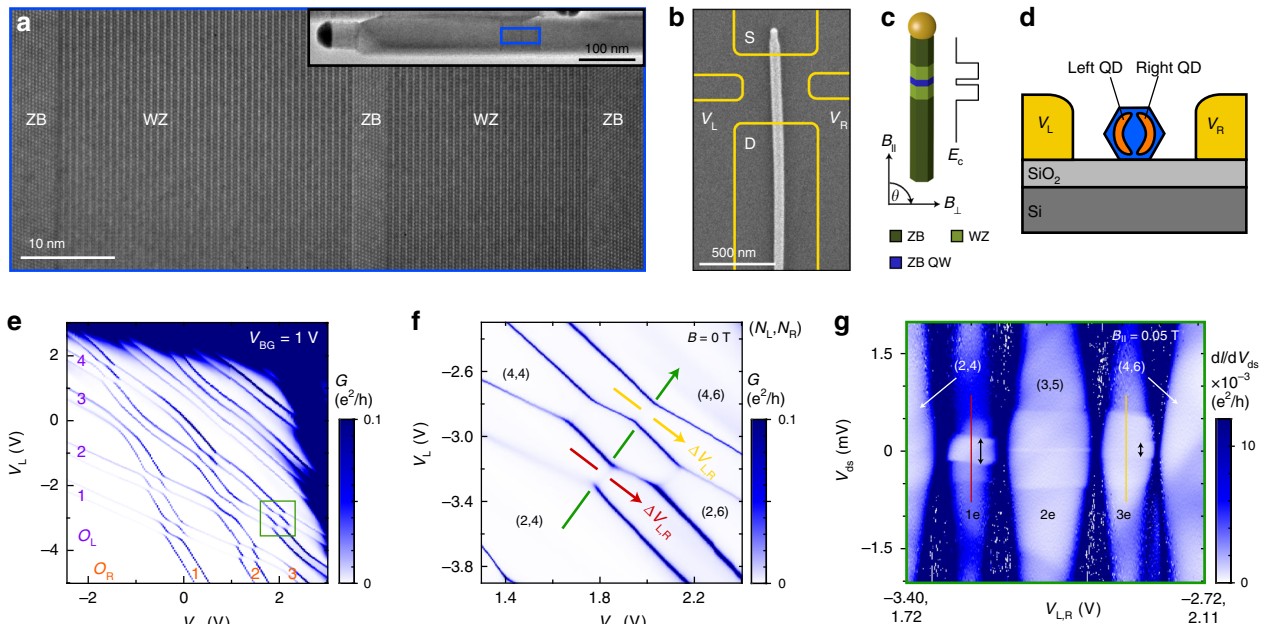

**Fig. 1 Formation of parallel-coupled QDs based on InAs nanowires. a** Transmission electron micrograph of a representative nanowire. The centre zinc blende (ZB) segment acts as a quantum well accessed by wurtzite (WZ) tunnel barriers. **b** SEM of the studied nanowire, overlaid with the contact design. **c** Schematic of the crystal structure and resulting conduction-band alignment. **d** Side-view illustration of the formation of two parallel-coupled QDs. **e** Conductance $G$ as a function of side-gate voltages ($V_L, V_R$), where the orbital numbers ($O_L, O_R$) are indicated. **f** Magnification of the ($O_L, O_R$) = (2,3) crossing. Green, red, and yellow arrows indicate important gate vectors, and ($N_L, N_R$) represents the electron population on the left and right QD. **g** Measurement of d$I$/d$V_{ds}$ versus $V_{ds}$ along the green gate vector for $B_{||} = 0.05$ T. The Zeeman splitting of the ground state $2E_Z$ is indicated with arrows in the 1e and 3e regime.

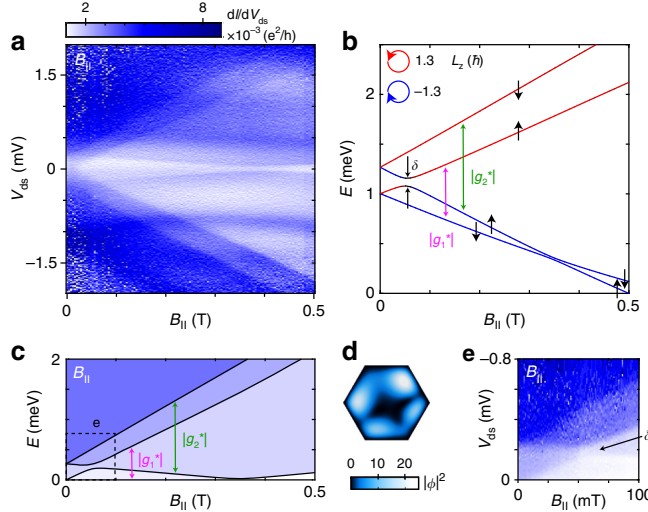

**Fig. 2 1e regime at zero detuning and $B_{||}$. a** Measurement of d$I$/d$V_{ds}$ versus $V_{ds}$ and $B_{||}$. **b** Numerical calculation of state energies as a function of $B_{||}$. The colour represents the calculated orbital angular momentum (blue: $L_z$ = −1.3 ℏ, black: $L_z$ = 0, red: $L_z$ = 1.3 ℏ). $|g_1^*|$ and $|g_2^*|$ correspond to the $B$-field induced splitting of the two Kramers pairs. The spin arrows indicate the alignment of the spin with respect to the nanowire axis (up/down corresponds to parallel/anti-parallel). Due to the presence of SOI, the arrows only indicate the approximate spin direction. **c** Calculated onsets of co-tunnelling processes starting from the lowest energy state as a function of $B_{||}$ (artificial colouring). **d** Absolute square of the wave function ($\Phi$) of the lowest energy state at infinitesimal $B_{||}$-field and $\Delta_{orb}$ = 0. **e** Same as panel **a**, zooming in on the small anti-crossing ($\delta$) due to coupling of states with same spin but different orbital momentum sign.

literature values. A nearly circular symmetry can also be identified in the calculated probability density distribution of the lowest energy state as shown in Fig. 2d. We obtain qualitatively similar results using a simple tight-binding model of a ring broken by two barriers (details of the models are provided in Supplementary Methods 1 and 2).

A finite $B_{||}$ couples to both the spin and the orbital momentum of the ring-like states, where $g^*$ is given by $g^* \approx g^*_{spin} \pm g^*_{orbit}$. Since the effective orbital g-factor ($g^*_{orbit}$) dominates, the two clockwise states decrease in energy with $B_{||}$, whereas the anti-clockwise states increase. For a perfect ring, the four states at $B$ = 0 are split by the spin-orbit interaction energy ($\Delta_{SOI}$) into two pairs, similar to the Kramers doublets (K↓, K'↑) and (K↑, K'↓) for carbon nanotubes. We now define $g_1^*$ and $g_2^*$ as the effective g-factor of the lower and higher Kramers doublet, respectively. A positive (negative) value for $g^*$ indicates that the lower-energy state is spin-down (spin-up). Experimental values of $g_1^* \approx 59$ and $g_2^* \approx -83$ are extracted from Fig. 2a, which roughly corresponds to $g^*_{spin} = -12$ and $g^*_{orbit} = 71$. We note that signs of $g_1^*$ and $g_2^*$ depend on the sign of SOI. In our case $g_1^*$ is positive as the (positive) orbital contribution is added to the (negative) spin-contribution, while $g_2^*$ is negative and larger in magnitude because both contributions have negative sign.

In a ring with disorder, states with different orbital momentum sign are coupled, resulting in an energy split ($\delta$) related to backscattering. The energy gap at $B$ = 0 is then given by $\Delta E_{B=0} = \sqrt{\Delta_{SOI}^2 + \delta^2}$. From the high resolution experimental data in Fig. 2e, we observe $\Delta E_{B=0} \approx 240$ μeV, and $\delta \approx 50$ μeV can be extracted from the anti-crossing at $B_{||} \approx 55$ mT. Interestingly, a comparable energy ($\delta_{KK'} \approx 65$ μeV) has been reported in ultra-clean carbon nanotube QDs, having a circumference of ~15 nm and device length of 500 nm[7], whereas our system has 250 nm

circumference and 5 nm length. The total length scales of the two systems are therefore comparable. Based on this $\delta$, we estimate $\Delta_{SOI} = 235\ \mu eV$, showing that $\Delta E_{B=0}$ is dominated by SOI, which confirms a high ring quality under optimum conditions. Going to even higher $B_{||}$, we ultimately find a change of the spin ground state at $B_{||} \approx 0.35$ T (Fig. 2a), where the Zeeman splitting of clockwise states (blue) overcomes the SOI-induced energy gap. Giant $g^\star$ involving GS-ES1 transitions are thus in our case accessible within an energy window of ~200 $\mu eV$, provided by the dimensions and SO-interaction of the system. This window can be enhanced using smaller ring diameters, and other materials, such as InSb.

Next, we investigate the evolution of states with $B_\perp$ applied perpendicular to the nanowire axis. With increasing $B_\perp$, a gap slowly opens between the ground state and the first excited state (ES1), as shown in Fig. 3a, b. The weak $B_\perp$ dependence is a result of the mixing of spin and orbital states due to SOI, since the avoided crossing strongly suppresses Zeeman splitting at small $B_\perp$[7]. Theory predicts that the GS-ES1 splitting converges to $\delta$ when both states have the same spin orientation at higher $B_\perp$. This strong non-linearity makes it difficult to define $g^\star$ of the ring-like state for $B_\perp$.

The exceptionally strong dependence of the state energies on the **B**-field direction ($\theta$) can be visualized by rotating **B** in the plane of the nanowire. An excellent agreement between experimental and corresponding simulation data is shown in Fig. 3c, d for $|\mathbf{B}| = 0.2\,$T. From the experiment we extract $|g_1^\star|$ and $|g_2^\star|$ as a function of **B**-field direction, plotted in Fig. 3e.

**Quenching the ring.** So far we have discussed the properties of high-quality quantum rings when two orbitals of different QDs align ($\Delta_{orb} = 0$). In the following, we investigate detuning of the involved orbitals with an electric field. This allows us to control the ring quality in situ, which has not been demonstrated for carbon nanotubes.

In Fig. 4a we present transport as a function of level detuning in the 1e regime (red vector in Fig. 1e) for $B_{||} = 0.1$ T. Detuning is here defined as the change in side-gate voltages with respect to orbital degeneracy ($V_{R0}$, $V_{L0}$), and can be converted to an energy using the leverarms (see Supplementary Fig. 1, Supplementary Table 1). Comparing with simulations (Fig. 4b), we see that the splitting between the ground state and the first excited state strongly depends on detuning and rapidly decays when going away from orbital degeneracy. This indicates that the ring state quenches with detuning.

The ability to control the magnetic field response ($g^\star$) using an electric field is important in many proposed concepts for spin manipulation, such as in spintronics and $g$-tensor modulation techniques[19,25,26]. Based on a linear approximation, we have extracted $|g^\star|$ from the GS-ES1 transition for $B_{||} = 0.1$ T, and $B_{||} = 0.04$ T (Fig. 4a and Supplementary Fig. 2e). The results in Fig. 4c show that $|g^\star|$ can be electrostatically tuned from above 80 down to almost 0, with a peak $dg^\star/d\Delta V_{L,R} > 1000$ V$^{-1}$, which is very attractive for $g$-tensor modulation.

In Fig. 4d we present a larger detuning range for a measurement at $B_{||} = 0.2$ T. By gradually destroying the quantum ring, and thereby decreasing its orbital contribution, the GS-ES1 energy gap approaches zero near $\Delta V_{R,L} = -0.2$ V, followed by a ground state change. This can be understood by considering that the ground state at orbital degeneracy is spin down (for $B_{||} < 0.35$ T), whereas the unperturbed orbital $O_L$ has a spin-up GS. Theory predicts that the ground-state change should show an exact crossing if the external **B**-field is aligned with the spin-orbit field (in this case spin is a good quantum number, regardless of disorder). A similar 1e spin ground state change was

demonstrated by Hauptmann et al. using electrostatic manipulation of the exchange field from ferromagnetic contacts to a carbon nanotube QD[27].

Next, we investigate the effect of detuning for the case of $B_\perp$. Figures 4e, f show the experimental data and corresponding simulation for $B_\perp = 0.1$ T. As previously discussed, mixing of spin and orbital states due to SOI leads to a suppression of the Zeeman splitting at orbital degeneracy. Near orbital degeneracy, a dip in the GS-ES1 energy gap is therefore observed both in the experimental and simulation data. Extracted values for $|g^\star|$ as functions of detuning are presented in Fig. 4g. The extracted $|g^\star| \sim 3$ at orbital degeneracy should be considered an upper bound as it is difficult to find the specific angle where no external field penetrates the ring. When the ring is quenched by detuning, $|g^\star|$ approaches 7.5, which is in line with typical values observed in InAs QDs[23,28]. The peak of $dg^\star/d\Delta V_{L,R}$ corresponds to >100 V$^{-1}$. The calculated probability density of the lowest energy state (Fig. 4h) shows a ring-like wave function at $\Delta_{orb} = 0$, whereas $\Delta_{orb} \neq 0$ shifts the wave function to either side and clearly destroys the ring symmetry.

**3-electron regime.** In the following, we discuss the 3e regime, which typically is equivalent to the 1e regime in spin-degenerate two-orbital systems due to particle-hole symmetry. However, this is not the case in ring-like QDs with strong SOI[7]. The specific ordering of the Kramers pairs is here determined by the signs of $\Delta_{SOI}$ and $g^\star_{spin}$. The orbital crossing (2,3) has a smaller splitting of the lower pair compared to the upper pair, which is consistent with $\Delta_{SOI} > 0$ and $g^\star_{spin} < 0$. This, however, is different for other

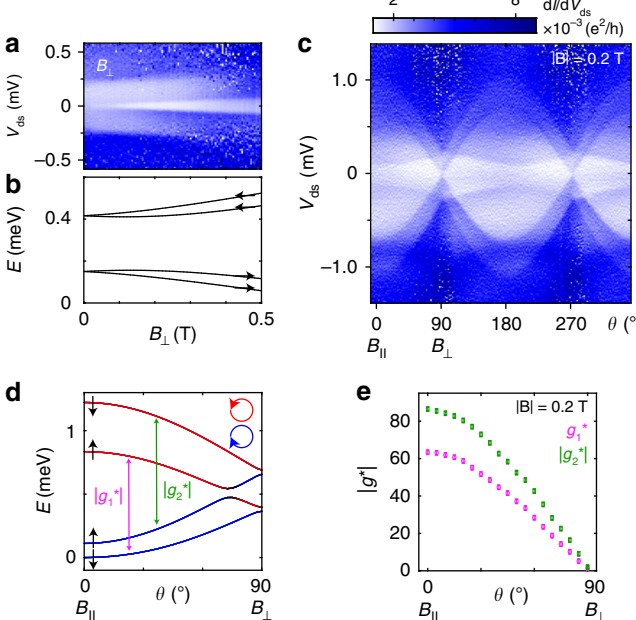

**Fig. 3 1e regime at zero detuning for different B-field directions. a** Measurement of $dI/dV_{ds}$ versus $V_{ds}$ and $B_\perp$. **b** Numerical calculation of the state energies as functions of $B_\perp$. The spin arrows indicate the alignment of the spin with respect to the external **B**-field. **c**, **d** Measurement of $dI/dV_{ds}$ and numerical calculation as a function of **B**-field direction for $|\mathbf{B}| = 0.2$ T. The colour represents the calculated orbital angular momentum (blue: $L_{z'} = -1.3\ \hbar$, black: $L_{z'} = 0$, red: $L_{z'} = 1.3\ \hbar$). The strong orientation dependence is a result of the orbital contribution to $g^\star$. **e** Experimental values of $|g_1^\star|$ and $|g_2^\star|$ as functions of **B**-field direction. The error bars represent the uncertainty in energy gap extraction due to measurement noise and resolution.

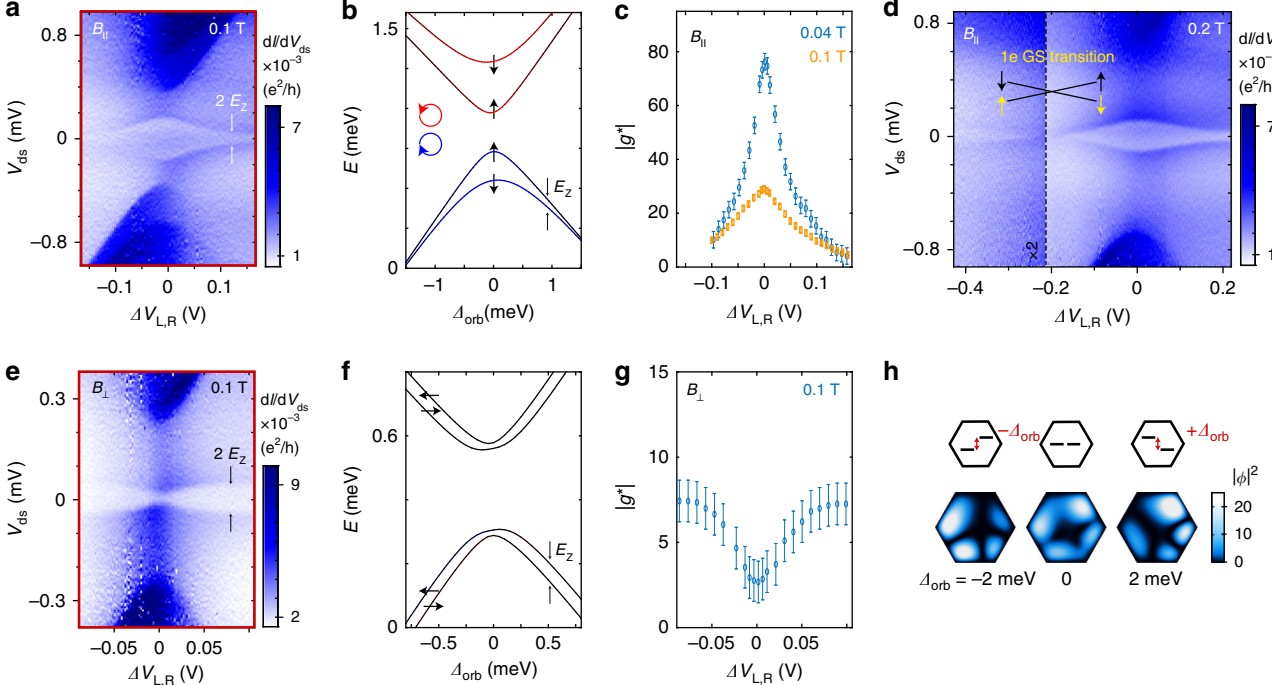

**Fig. 4 Quenching the 1e quantum ring by detuning. a** Measurement of $dI/dV_{ds}$ versus $V_{ds}$ recorded along the red detuning vector (c.f. Fig. 1f) for $B_∥ = 0.1$ T. Detuning is defined as $\Delta V_{L,R} = (V_R − V_{RO}) − (V_L − V_{LO})$, where $(V_{RO}, V_{LO})$ are the side-gate voltages when the left and right QD levels are degenerate. $E_Z$ denotes the Zeeman splitting. **b** Numerical calculation of the states for $B_∥ = 0.1$ T. Here detuning ($\Delta_{orb}$) refers to the extrapolated energy difference between the unperturbed orbitals in the left and right QD (see schematic in **h**). The colour represents the calculated orbital angular momentum (blue: $L_z = -1.3\ \hbar$, black: $L_z = 0$, red: $L_z = 1.3\ \hbar$). **c** Experimental values of $|g^*|$ for $B_∥ = 0.04$ T (blue) and $B_∥ = 0.1$ T (orange), extracted from the GS-ES1 transition using on a linear approximation. In the case of $B_∥ = 0.04$ T, $|g^*|$ at zero detuning corresponds to $|g_1^*|$, while we find a strongly reduced value at $B_∥ = 0.1$ T due to the orbital change of the 1st excited state. **d** Same as panel a, but for $B_∥ = 0.2$ T and a larger detuning range, showing a detuning-induced spin-change of the 1e ground state. **e, f** Measurement of $dI/dV_{ds}$ and numerical calculation as a function of detuning for $B_⊥ = 0.1$ T, where SOI from the ring suppresses Zeeman splitting. **g** Experimental values of $g^*$ for the GS-ES1 transition at $B_⊥ = 0.1$ T. **h** Calculated probability density ($|\Phi|^2$) of the lowest energy state at infinitesimal $B_∥$-field for three different detuning energies. The error bars in **c** and **g** represent the uncertainty in energy gap extraction due to measurement noise and resolution.

orbital crossings, as shown in Supplementary Figs. 5 and 7, where $\Delta_{SOI} < 0$ is found.

The 3e case can be understood by considering that the first three states are filled, and transitions to the 4th state are probed. Opposite to the 1e case, the GS-ES1 energy gap at orbital degeneracy should therefore continuously increase with $B_∥$. Spectroscopic data recorded as a function of $B_∥$ (Fig. 5a) and **B**-field angle (Fig. 5b) agrees with this interpretation. Figure 5c shows the characteristic increase of the ES1-GS energy upon formation of the ring-like state. More experimental data for both $B_∥$ and $B_⊥$ can be found in Supplementary Fig. 3. Figure 5d shows $|g^*|$ extracted as a function of detuning for the 3e regime. Similar to the 1e case, $|g^*|$ recorded for $B_∥ < 55$ mT can directly be interpreted as $|g_2^*|$. The maximum value of $|g_2^*| \approx 83$ at orbital degeneracy matches with the estimation based on the $B_∥$-sweep of the 1e case. However, as the 4th state is already spin-down at orbital degeneracy, no change in ground state spin is observed with detuning, which also reduces the peak $dg^*/d\Delta V_{L,R} > 800$ V$^{-1}$ compared to the 1e case.

**Orbital requirements for ring formation.** Finally, we address the general question of the requirements necessary to observe these ring-like states. Based on our tight-binding model and a simpler DQD model (details in Supplementary Methods 2) we find that a nearly perfect ring can form even in the presence of significant tunnel barriers between the QDs if two conditions are fulfilled: (1) the tunnel coupling strength at the two connection points of the QDs are identical, and (2) an even orbital in one QD is

energetically aligned with an odd orbital in the other QD. An even/even or odd/odd combination of orbitals (first two panels in Fig. 5e) leads to a large energy gap between the lower and the higher Kramers pair, similar to standard bonding and anti-bonding orbitals of QDs, which are connected in one point. However, in the case of an even/odd combination (third panel of Fig. 5e), the hybridization gap vanishes because of the different signs of the overlap integrals ($S$) at the two QD connection points. The resulting four degenerate states now split in the presence of SOI into two Kramers pairs, each of which show ring-like behaviour. There is thus no strict requirement for a particular material, and other coupled QD systems should display similar physics. This may extend the possibilities for spin manipulation and tuning of $g^*$ in QD systems with longer spin coherence times, and allow for strong coupling of spins to cavity photons.

## Methods

**Nanowire growth.** InAs nanowires with controlled crystal structure were grown by metal-organic vapour phase epitaxy (MOVPE) in an AIXTRON $3 \times 2''$ close coupled showerhead reactor. Before growth, arrays of Au pads were defined on InAs 111B substrates by electron beam lithography followed by thermal evaporation of Au. After loading the substrates into the MOVPE reactor, an annealing step at a set temperature of 550 °C was carried out for 7 min in an AsH$_3$/H$_2$ atmosphere with an AsH$_3$ molar fraction of $\chi_{AsH3} = 2.5E-3$ before ramping down to the set growth temperature of 460 °C. Nanowire growth was initiated by introducing tri-methylindium (TMIn) at a molar fraction of $\chi_{TMIn} = 2.7E-6$ and lowering the AsH$_3$ molar fraction to $\chi_{AsH3} = 1.2E-4$ for a stem growth of 240 s. The crystal phase switching was realised by changing the AsH$_3$ molar fractions from $\chi_{AsH3} = 2.5E-2$ for ZB growth to $\chi_{AsH3} = 2.2E-5$ for WZ growth, with 15 s waiting steps under AsH$_3$. The growth time was 360 and 480 s for the bottom and top ZB segments,

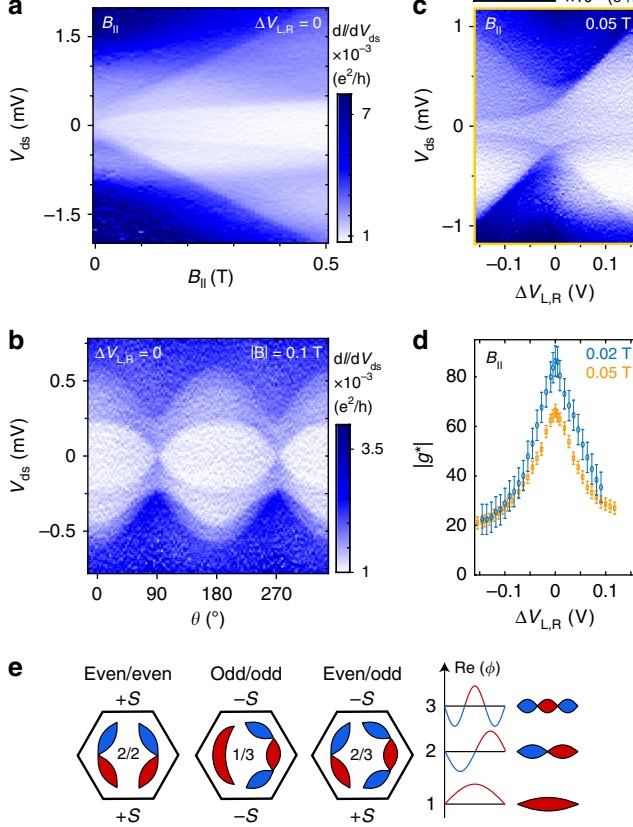

**Fig. 5 3e regime and requirements for ring formation. a** Measurement of $dI/dV_{ds}$ versus $V_{ds}$ and $B_{\parallel}$ at zero detuning. The SOI breaks e-h symmetry, leading to an increasing GS-ES gap with $B_{\parallel}$ for all three excited states in the 3e regime. **b** Measurement of $dI/dV_{ds}$ as a function of **B**-field direction for $|\mathbf{B}| = 0.1\,\text{T}$ at zero detuning. **c** Measurement of $dI/dV_{ds}$ when detuning the orbitals along the yellow gate vector (c.f. Fig. 1f) for $B_{\parallel} = 0.05\,\text{T}$. **d** Experimental values of $|g^*|$ for the GS-ES1 transition based on a linear approximation. In the case of $B_{\parallel} = 0.02\,\text{T}$ (blue), the extracted $|g^*|$ corresponds to $|g_2^*|$, while we find a reduced value at $B_{\parallel} = 0.05\,\text{T}$ (orange) due to the avoided orbital crossing near this $B_{\parallel}$-field. The error bars represent the uncertainty in energy gap extraction due to measurement noise and resolution. **e** Schematic representation of the real part of the wave functions (Re($\Phi$)) in crossings of even/even (2/2), odd/odd (1/1), and even/odd (2/3) orbitals. The blue and red colour correspond to negative and positive Re($\Phi$). Nearly perfect rings can only be observed in the case of even/odd combinations where the overlap integral ($S$) has different sign at the two connection points.

respectively, and 18 and 10 s for the WZ barriers and ZB quantum well, respectively. After the axial growth of the alternating crystal phase segments was finished, the sample was cooled down to a set temperature of 300 °C in an $AsH_3/H_2$ atmosphere with an $AsH_3$ molar fraction of $\chi_{AsH3} = 2.5E\text{-}3$. The crystal structure of the nanowires was analysed in a Hitachi HF3300S transmission electron microscope operated at 300 keV.

**Device fabrication**. Select nanowires were deposited with a micromanipulator onto highly doped silicon samples with 200 nm $SiO_2$. Contacts were then fabricated using e-beam lithography followed by native oxide removal in HCL(37%):$H_2O$ (1:20) for 15 s and evaporation of 20/80 nm Ni/Au.

**Transport measurements**. All experimental data are obtained from asymmetrically biased DC measurements performed in a dilution refrigerator at an electron temperature <100 mK.

**Data analysis**. The differential conductance was calculated using the gradient function in Matlab. In some cases, noise from the raw data was reduced by a moving average (smooth function) or a local regression (rlowess function). The values of $g^*$ are determined based on a linear approximation. The $B_{\parallel}$-field dependence (Fig. 2a) allows an accurate extraction of $g_1^*$ and $g_2^*$, by taking data points, which are far away from the hybridization gaps. In the case of the detuning dependence (Fig. 4c, g and 5d), the reported $g^*$ corresponds to the energy gap between the ground state and first excited state, and is therefore influenced by the hybridization of states. The error bars of $g^*$ are calculated based on the resolution of $V_{ds}$ (10 μV for Fig. 3e, and 5 μV for Fig. 4c, g and 5d).

**Theoretical models**. The theoretical analysis presented in the main article is based on a three-dimensional numerical simulation of single-electron states using COMSOL. A detailed description of the model can be found in the Supplementary Methods 1 (Supplementary Figs. 8 and 9, Supplementary Tables 2 and 3). Additionally, we have performed tight-binding calculations of a ring broken by two barriers. The results from the tight-binding calculations, as well as a detailed description of the model can be found in the Supplementary Methods 2 (Supplementary Figs. 10 and 11).

## Data availability
The data that support the findings of this study are available from the corresponding authors upon request.

## Code availability
The code of the simulations is available from the corresponding authors upon request.

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

## Acknowledgements

The authors thank A. Burke for experimental support. Financial support was received from the Knut and Alice Wallenberg Foundation (KAW), the Swedish Research Council (VR), the Crafoord Foundation and NanoLund. H.P. acknowledges the Swiss National Science Foundation (SNSF) via Early PostDoc Mobility.

## Author contributions

H.P., I-J.C. and C.T. did the transport measurements, analysis and figure preparation. I.-J.C., A.T. and M.L. developed the theoretical models. H.P. and M.N. fabricated the samples. S.L. and K.A.D. did the epitaxy and structural characterization. All authors contributed to discussions and writing of the manuscript.

## Competing intersts

The authors declare no competing interests.
