## [Peer Review File · Nature Communications]

Reviewers' comments:

Reviewer #1 (Remarks to the Author):

referee report on manuscript "Electrical control of spins and giant g-factors in ring-like coupled quantum dots" by Potts et al.

The manuscript reports about experiments and calculations on quantum dots defined by crystal phases in InAs nanowires. The authors find strong orbital contributions to the splitting in magnetic field and explain the observations by ring-like dot states, which can be tuned by electric fields. In analogy to carbon nanotubes such pronounced spin-orbit interaction leads to an energy spectrum which strongly depends on the direction of the magnetic field.

The experiments are expertly done and the interpretation is sound. While many quantum dot spectroscopy experiments have been done in general and using InAs nanowire quantum dots in particular, this manuscript adds an important new class of quantum dots that may lend themselves for the use in quantum information processing.

I enthusiastically recommend to publish this manuscript in Nature Comm.

Reviewer #2 (Remarks to the Author):

The article reports on g-factor measurements of electron states localized in an InAs nanowire quantum dot. The quantum dot is embedded in the nanowire during the growth and can be thus viewed as 2DEG created on the cross-section of the nanowire.

The article focus is on transport spectroscopy, providing information on the orbital and spin structure of the single electron states of the quantum dot. It confirms the basic picture of a two-dimensional dot with highly tunable (by side gates) shape, which, together with the strong spin-orbit field of InAs leads to large g-factor anisotropy.

I find this to be an exciting platform, with potential for several current subfields of semiconductor nanoscopic physics. The analysis is solid, measurements supported by simulations, and the

presentation is appropriate. I am therefore supportive of a publication after authors address the following issues.

1. I do not understand the motivation of extensive comparison to a carbon nanotube in the introduction. For example, Ref. 6 dealt with a nanotube with 0.5 μm length, while here a 5 nm portion of a "wire" only is probed. I, therefore, find the comments on the orbital moment preservation being better than in "an ultra-clean" nanotube misleading. As far as I know, the InAs wires are typically dirty, e.g., not showing a nice conductance quantization. Do the authors suggest by those comments that these InAs wire are different, exceptionally clean?

2. I wonder about the usability of the exceptionally large g -factor. First, it is large only if the external field is along the nanowire axis. It is not clear to me where the spin-orbit field points to (see point 3 below). Second, the achievable "Zeeman" gap is limited by the spin-orbit gap (minus $\Delta/2$) being about 200 μeV . So, in this respect, it is the spin-orbit gap rather than g -factor itself which is the most important parameter (GaAs would have comparable g -factor within the same two states subspace, but a minute achievable maximal "Zeeman" energy). Could the authors comment on that?

3. Why a Rashba spin-orbit interaction generated by an E_z field was assumed? The structure looks symmetric along the growth axis to me. If there is any Rashba, it should be produced by the surface charges, with the field pointing into the nanowire. Alternatively, a bulk Dresselhaus interaction should be considered. Will the conclusions remain the same? Especially on the spin direction of the two states?

4. When fitting the data to models, effective mass gives Rashba constant of 16 meVnm , while the tight binding gives 2 meVnm . What is the meaning of the large discrepancy?

5. The expectation value of the orbital momentum L_z is fitted in SI page 6 to ~ 1 . Is this a measure of the transparency between the left-right states? Would transparency 1 mean $L_z=5$ or $L_z=1$ (or something else) in this configuration?

6. "B-dependence of the overview" in SI, page 13, is what?

7. Finally, I urge the authors to improve the resolution of the figures in the main text. The labels (especially subscripts) are close to unreadable when printed, as well as the markings (arrows). SI does not have this problem, so the solution should be possible.

Reviewer #3 (Remarks to the Author):

Ref: NCOMMS-19-18764-T

The authors presented an experimental study of giant g-factors in quantum dots formed in InAs nanowires supplemented by a numerical simulation. Compared with previous experimental work [Nature 531, 206 (2016)] and theoretical explanation [PRL 119, 037701 (2017)], this submission provides another insight into giant g-factors in nanowires and nanodots and shows its potential application in spintronics. However, the improvement over those works is believed to be only quantitative instead of qualitative. Therefore, it may be suitable for publication in more specialized or applied journals.

Reviewer #1:

The manuscript reports about experiments and calculations on quantum dots defined by crystal phases in InAs nanowires. The authors find strong orbital contributions to the splitting in magnetic field and explain the observations by ring-like dot states, which can be tuned by electric fields. In analogy to carbon nanotubes such pronounced spin-orbit interaction leads to an energy spectrum which strongly depends on the direction of the magnetic field.

The experiments are expertly done and the interpretation is sound. While many quantum dot spectroscopy experiments have been done in general and using InAs nanowire quantum dots in particular, this manuscript adds an important new class of quantum dots that may lend themselves for the use in quantum information processing.

I enthusiastically recommend to publish this manuscript in Nature Comm.

We thank the referee for the very positive feedback!

Reviewer #2:

The article reports on g -factor measurements of electron states localized in an InAs nanowire quantum dot. The quantum dot is embedded in the nanowire during the growth and can be thus viewed as 2DEG created on the cross-section of the nanowire.

The article focus is on transport spectroscopy, providing information on the orbital and spin structure of the single electron states of the quantum dot. It confirms the basic picture of a two-dimensional dot with highly tunable (by side gates) shape, which, together with the strong spin-orbit field of InAs leads to large g -factor anisotropy.

I find this to be an exciting platform, with potential for several current subfields of semiconductor nanoscopic physics. The analysis is solid, measurements supported by simulations, and the presentation is appropriate. I am therefore supportive of a publication after authors address the following issues.

We thank the referee for the positive feedback and appreciate the opportunity to clarify a few points.

1. I do not understand the motivation of extensive comparison to a carbon nanotube in the introduction. For example, Ref. 6 dealt with a nanotube with 0.5 μm length, while here a 5 nm portion of a "wire" only is probed. I, therefore, find the comments on the orbital moment preservation being better than in "an ultra-clean" nanotube misleading. As far as I know, the InAs wires are typically dirty, e.g., not showing a nice conductance quantization. Do the authors suggest by those comments that these InAs wire are different, exceptionally clean?

We motivate the comparison by the ring-like architecture of both QD systems, leading to a very similar energy spectrum. We note that orbital momentum conservation is limited by all three dimensions of these systems. In the CNT experiments, the circumference is small, ca 15 nm, but the segment is long, 500 nm. In our case the segment is short, ca 5 nm, but the circumference is large, about 250 nm. The total length scales of the two systems are therefore comparable.

However, we do agree that our system is likely much less clean than the CNT case, and this is where the important difference comes in: The orbital conservation that we see only occurs under certain, reproducible conditions, which is key to the g -factor tuning. The localized QD states themselves typically have no (or very small) orbital moment whereas the extended ring-like states that form under resonance conditions have a large orbital momentum, despite a potentially dirty underlying system. In fact, this sensitive nature of the orbital moment in our system is extremely useful as it allows a rather small gate voltage to dramatically change the g -factor, which is not possible in a carbon nanotube.

We have now emphasized “under optimum ring conditions” when comparing our results with the results from CNTs. This is to make it more clear that the underlying system is different, and that the parameter space where similar values are found is limited. In the discussion around Figure 2, we have also added some additional information about the system dimensions when we compare to the CNT case.

The following section in the introduction was restructured, and new text was added (in blue):

Notably, the electronic structure of the ring-like states is almost identical to that of carbon nanotube QDs, having a nearly fourfold orbital- and spin-degeneracy, broken by spin-orbit interaction [6, 17, 18]. However, in contrast to carbon nanotubes, we can dramatically affect the electron wave function by forming or quenching the ring-like orbitals using electric fields. As a result, we show that $|g^*|$ can be electrostatically controlled from ~ 80 to ~ 0 for the same charge state, and the ground state spin can be changed in a constant magnetic field. Previous reports on tuning g^* in QDs cover a much smaller span of modulation [19-21] and often rely on changing the charge state [8, 9]. The quality of the rings can be quantified by a back-scattering term (δ) connecting states with opposite orbital momentum sign. Under optimum ring conditions, we find a value of δ comparable to ultra-clean carbon nanotubes [6].

Updated and new text in the discussion of Figure 2:

From the high resolution experimental data in Fig. 2e, we observe $\Delta E_{B=0} \approx 240 \mu\text{eV}$, and $\delta \approx 50 \mu\text{eV}$ can be extracted from the anti-crossing at $B_{\parallel} \approx 55 \text{ mT}$. Interestingly, a comparable energy ($\delta_{\text{KK}'} \approx 65 \mu\text{eV}$) has been reported in ultra-clean carbon nanotube QDs, having a circumference of approximately 15 nm and device length of 500 nm [6], whereas our system has 250 nm circumference and 5 nm length. The total length scales of the two systems are therefore comparable. Based on this δ , we estimate $\Delta_{\text{SOI}} = 235 \mu\text{eV}$, showing that $\Delta E_{B=0}$ is dominated by SOI, which confirms a high ring quality under optimum conditions.

2. I wonder about the usability of the exceptionally large g -factor. First, it is large only if the external field is along the nanowire axis. It is not clear to me where the spin-orbit field points to (see point 3 below). Second, the achievable "Zeeman" gap is limited by the spin-orbit gap (minus $\delta/2$) being about 200 μeV . So, in this respect, it is the spin-orbit gap rather than g -factor itself which is the most important parameter (GaAs would have comparable g -factor within the same two states subspace, but a minute achievable maximal "Zeeman" energy). Could the authors comment on that?

We agree that it is relevant to discuss the usability in a bit more detail. As the referee points out, the GS-ES1 excitations in the same orbital are limited by the spin-orbit gap (in the one-electron regime), indeed making InAs a more favourable material than GaAs. Although the energies are limited to some hundred μeV , the g -factor magnitude itself is however still very important for some studies and applications involving small energy scales and/or where the B -field needs to be limited. This could be in studies of hybrid semiconducting/superconducting systems (Majorana), or for integration with superconducting resonator structures. For electrically driven spin resonance studies (spin qubits), it is desirable to be able to tune the g -factor with an electric field, in addition to a large g -factor magnitude. If the sensitivity to the electric field is very high (which we consider the most important point of our manuscript), then it is possible to use weak magnetic fields (Ares *et al*, Ref 20).

We note that the electrical g -factor tuning also works for transverse B -fields and at much higher B -fields and energies. However, the g -factor is here suppressed as we point out in the manuscript.

We have added the following two sentences to the manuscript:

Giant g^* involving GS-ES1 transitions are thus in our case accessible within an energy window of $\sim 200 \mu\text{eV}$, provided by the dimensions and SO-interaction of the system. This window can be enhanced using smaller ring diameters, and other materials, such as InSb.

3. Why a Rashba spin-orbit interaction generated by an E_z field was assumed? The structure looks symmetric along the growth axis to me. If there is any Rashba, it should be produced by the surface charges, with the field pointing into the nanowire. Alternatively, a bulk Dresselhaus interaction should be considered. Will the conclusions remain the same? Especially on the spin direction of the two states?

An electric field in the NW axial direction can be justified by considering polarization effects at the zinc-blende/wurtzite junctions (spontaneous or strain-related). The continuum model calculations were performed both with a SOI generated by a radial E -field and with a field along the z -direction, although only the latter was included in the manuscript. We found that both choices can give very similar energy spectra, and also the spin structure away from zero field remains similar. We have provided more information in the SI (Table S1), see also question 4.

In the tight-binding model a Rashba SOI generated by a radial E -field, as suggested by the referee, was indeed used.

Concerning the experimental results, if the electrical field was purely radial, we would have expected to see an exact crossing in Figure 2a when the spin ground state undergoes a change at $B_{||} \approx 0.35 \text{ T}$, but instead we see a small avoided crossing.

4. When fitting the data to models, effective mass gives Rashba constant of 16 meVnm , while the tight binding gives 2 meVnm . What is the meaning of the large discrepancy?

We thank the referee for pointing out that the comparison of the models would benefit from additional clarification. In the submitted manuscript, the two values of the Rashba constant cannot directly be compared because they were obtained based on different directions of the spin-orbit field

(a radial field in the case of the tight binding model, and an axial field for the 3D simulation). In order to compare the two models quantitatively, we have now provided more data about the 3D simulation with a radial spin-orbit field in the revised supplementary file (see updated text and Table S1 in the supplementary). In the 3D simulation with radial spin-orbit field, the Rashba constant depends on the location of the electrons and is given by $\alpha^* = (\beta_R^x, \beta_R^y, 0)$, where x, y are the coordinates of the electron and R is the nanowire diameter. A good agreement with the experiment is obtained for $\beta = 4.2$ meVnm. This value is an upper bound for the Rashba parameter since the electrons are not located directly at the nanowire surface ($\sqrt{x^2 + y^2} < R$, see calculated probability density in Fig. 4h). Comparing the 3D where a radial field was used to the tight binding model, the difference in Rashba constant is therefore less than a factor of two. For the 3D simulation with axial spin-orbit field, we use a value of 16 meVnm to get good agreement with the experiment, however this is within the range of values found in literature (which extend up to 30 in the paper by Winkler *et al*, Ref 16).

5. The expectation value of the orbital momentum L_z is fitted in SI page 6 to ~ 1 . Is this a measure of the transparency between the left-right states? Would transparency 1 mean $L_z=5$ or $L_z=1$ (or something else) in this configuration?

The L_z result is the expectation value ($/\hbar$) of the orbital momentum operator. Our (rather remarkable) finding is that ring-like states form in (even, odd) crossings even though the transparency is low. However, if the transparency is close to 1, the concept of a (2,3) crossing (for example) is not really meaningful, as this would mean that there is no barrier and therefore no way to define left and right dots or orbitals.

However, we do note that a perfect ring with 7 electrons would have $L = 2$ for the 7th electron. This number ($L = 2$) can be compared with the values of $L = 1.3$ (3D) and $L = 1.1$ (tight-binding) we find in the models for a QD-ring system with 7 electrons in the (2,3) crossing.

We have now added the following statement to the S.I. on page 6:

We note that a perfect ring with 7 electrons, corresponding to the 1e regime of the (2,3) crossing in terms of electron number, would have $L_z = 2 \hbar$ for the electron in the outermost populated orbital.

6. "B-dependence of the overview" in SI, page 13, is what?

We agree that the terminology in this section can be improved. We have changed the section heading to '*B-field dependence of the honeycomb diagram*', and made a few changes to the text. In summary, this section shows that by plotting the conductance as a function of side-gate voltages for different B -field strengths and orientations, we can easily visualize that the hybridization of the QD states depends on the B -field. Very sharp corners are observed in the honeycomb diagram for zero B -field or for perpendicular B -field, while a parallel B -field leads to round corners. Such a B -field dependence of the ground state only occurs for crossings with quantum ring states, allowing to identify ring-like states very efficiently.

7. Finally, I urge the authors to improve the resolution of the figures in the main text. The labels (especially subscripts) are close to unreadable when printed, as well as the markings (arrows). SI does not have this problem, so the solution should be possible.

We apologize for the inconvenience. The figures of the main article and the SI were prepared with the same resolution, the problem in the main article therefore must have occurred during the conversion on the submission platform. For the final version of the article we will submit all figures as vector graph files, and thereby ensure good resolution of text and markings.

Reviewer #3:

The authors presented an experimental study of giant g -factors in quantum dots formed in InAs nanowires supplemented by a numerical simulation. Compared with previous experimental work [Nature 531, 206 (2016)] and theoretical explanation [PRL 119, 037701 (2017)], this submission provides another insight into giant g -factors in nanowires and nanodots and shows its potential application in spintronics. However, the improvement over those works is believed to be only quantitative instead of qualitative. Therefore, it may be suitable for publication in more specialized or applied journals.

We thank the referee for her/his comment. We are aware that giant g -factors have been investigated in the works cited by the referee, and the two articles are also cited in several places in our manuscript. However, we would like to stress that our work focuses on the tunability of the g -factor with small electric fields. This is only possible due to the unique architecture of our system, where two quantum dots can be coupled in order to form a ring under certain conditions. A small deviation from the ideal conditions leads to a strong suppression of the g -factor, allowing very efficient tuning of the g -factor by small electric fields. This behaviour, which we observe experimentally and explain theoretically, is qualitatively significantly different from any existing work. The relevant phrases in the abstract/introduction are highlighted below:

Abstract:

By hybridizing specific quantum dot states at two points inside InAs nanowires, nearly perfect quantum rings form.

...

Importantly, we find that the orbital and spin-orbital contributions can be efficiently quenched by simply detuning the individual quantum dot levels with an electric field. In this way, we demonstrate not only control of the effective g -factor from 80 to almost 0 for the same charge state, but also electrostatic change of the ground state spin.

Introduction:

In this work, we explore the physics of strongly confined QDs coupled in a new geometry, having two connection points instead of just one.

...

However, in contrast to carbon nanotubes, we can dramatically affect the electron wave function by forming or quenching the ring-like orbitals using electric fields. As a result, we show that $|g^*|$ can be electrostatically controlled from ~ 80 to ~ 0 for the same charge state, and the ground state spin can be changed in a constant magnetic field. Previous reports on tuning g^* in QDs cover a much smaller span of modulation [19-21] and often rely on changing the charge state [8, 9].

...

We develop theoretical models showing excellent agreement with the experimental data. Using perturbation theory, we furthermore find that the contribution from the inter-dot tunnel-coupling to the hybridization gap can effectively cancel if an odd orbital on one QD hybridizes with an even orbital on the other QD. This method of creating high quality quantum rings is thus generic, and opens up for g^* manipulation and spin control in many material systems.

REVIEWERS' COMMENTS:

Reviewer #2 (Remarks to the Author):

I have reviewed the authors' reply and the resubmission version of the manuscript. I find all the issues raised by me, and also by the third referee, to be fully resolved and I recommend to publish the manuscript as it is.

Reviewer #3 (Remarks to the Author):

Having read the responses to mine and other referees' comments, I would recommend its publication. The only hesitation I have right now is that the unique coupled quantum dots in a ring-like structure seem unlikely to be reproduced by other researchers and may hence limit the influence of this work in the field.

REVIEWERS' COMMENTS:

Reviewer #2 (Remarks to the Author):

I have reviewed the authors' reply and the resubmission version of the manuscript. I find all the issues raised by me, and also by the third referee, to be fully resolved and I recommend to publish the manuscript as it is.

We thank the referee for the positive review and recommendation.

Reviewer #3 (Remarks to the Author):

Having read the responses to mine and other referees' comments, I would recommend its publication. The only hesitation I have right now is that the unique coupled quantum dots in a ring-like structure seem unlikely to be reproduced by other researchers and may hence limit the influence of this work in the field.

We thank the referee for the positive review and recommendation.